# Optimization of the Mechanical Recycling of Phenolic Resins for Household Appliances

**DOI:** 10.3390/polym16233378

**Published:** 2024-11-30

**Authors:** Francesco Valentini, Daniele Rigotti, Matteo Saletti, Alberto Beccaro, Laura Pasquardini, Alessandro Pegoretti, Andrea Dorigato

**Affiliations:** 1Department of Industrial Engineering and INSTM Research Unit, University of Trento, Via Sommarive 9, 38123 Trento, Italy; daniele.rigotti@unitn.it (D.R.); alessandro.pegoretti@unitn.it (A.P.); andrea.dorigato@unitn.it (A.D.); 2Nuova Saimpa srl, Loc. al Ponte, 38083 Borgo Chiese, Italy; saletti.matteo@nuovasaimpa.it (M.S.); alberto.beccaro@nuovasaimpa.it (A.B.); 3Indivenire srl, Via Sommarive 18, 38123 Trento, Italy; l.pasquardini@indiveni.re; 4Department of Engineering, University of Campania “Luigi Vanvitelli”, Via Roma 29, 81031 Aversa, Italy

**Keywords:** phenolic resin, resol, novolac, recycling, life cycle assessment

## Abstract

In light of the significant impact of climate change, it is imperative to identify effective solutions to reduce the environmental burdens of industrial production and to promote recycling strategies also for thermosetting polymers. In this work, the mechanical recycling of phenolic resins, obtained from industrial production scrap of plastic knobs for household appliances, was optimized. The feasibility of a partial substitution of virgin materials with recycled ones was investigated both at a laboratory and industrial scale. Finally, the environmental benefits arising from the use of recycled material were quantified through a life cycle assessment (LCA). The results of laboratory characterization demonstrated that the thermal properties of the phenolic resins were not influenced by the presence of recycled material, and the mechanical performances were not significantly impaired up to a recycled content of 30 wt%. The industrial production trials demonstrated the feasibility of replacing up to 15 wt% of virgin material without any influence on the aesthetical features of the produced components. Finally, LCA of industrially produced knobs highlighted a limited benefit of virgin material substitution in the case of novolac chromium-plated samples, while an overall environmental impact reduction of around 7–10% was detected in the case of resol-based materials.

## 1. Introduction

Phenolic resins are thermosetting polymers with unique properties that are widely used worldwide and whose consumption in 2014 was around 3.7 Mton, with an expected annual growth rate higher than 5% until 2028 [1,2]. The phenol-formaldehyde resin was first discovered in 1872 by the German chemist Adolf Von Baeyer (Nobel Prize in 1905) but industrial production was developed in 1910 by Leo Hendrik Baekeland with the foundation of the “General Bakelite Company” [3,4]. Due to their unique chemical structure and macromolecular cross-linked structure, phenolic resins exhibit excellent mechanical properties, flame retardancy, chemical and thermal stability, and aesthetical properties, being widely used for the production of electrical and electronic components, laminates, wood and fibre-reinforced composites, automotive parts and insulating foams [3,5]. The production of phenolic resins can be performed through a polycondensation reaction between phenol (P) and formaldehyde (F): when the molar ratio F/P is higher than 1 (typically 1.6–2.6) resol-type resins are obtained while when the F/P ratio is lower than 1 (typically 0.75–0.85) novolac-type resins can be synthesized [6,7]. The final properties of phenolic resins strongly depend on several factors, such as reaction temperature and time, pH, catalyst type and amount and starting F/P molar ratio [8,9,10,11,12]. As for the other thermosetting polymers, the cross-linked nature of phenolic resins makes them insoluble and infusible with consequent difficulties in the recycling process [13,14,15]. Despite the unavailability of official end-of-life statistics of phenolic resins, in the literature, it has been reported that the most common disposal route of this material is landfilling [16,17], while other options are pyrolysis [18] or its use in kilns or as inert material in concrete and asphalt [16,17]. Some attempts regarding the possibility of recycling phenolic resins are reported in the literature and can be divided between mechanical and chemical recycling.

Mechanical recycling is generally performed through high-speed grinding for a long time at high shear stresses and friction values, in order to partially break the chemical bonds and thus the crosslinked structure [19,20,21,22]. The resulting material is characterized by low quality due to the partial thermal decomposition that occurred during the recycling process and has a low practical value [17]. Moreover, in case of the presence of free phenol or free formaldehyde within the phenolic resin, they can be released during the mechanical recycling process [23]. Possible applications of the recycled material regard its incorporation within thermoplastic matrices as filler [24], its mixing with wood fibres for the preparation of panels with enhanced fire resistance [25], or its use as an antioxidant in the production of polypropylene and polyamide-6 [23]. Chemical recycling, on the other hand, involves the depolymerization of the phenolic resin or its conversion to carbon materials [17]. The main problems are the high amounts of energy required in the process and that the properties of carbon materials obtained from the conversion process are unstable [17].

Considering the absence of a recycling chain for phenolic resins and the limited number of studies present in the literature regarding their recycling, this study aims at the optimization of a mechanical recycling process of an industrial production scrap obtained from the production of knobs for household appliances. In particular, laboratory tests were performed to optimize the recycling process and to characterize the resulting materials. The optimized recycling procedure was then validated at an industrial level with the production of knobs containing a fixed amount of recycled resin. Finally, the environmental benefits arising from the use of recycled material were quantified through a comparative life cycle assessment.

## 2. Materials and Methods

### 2.1. Materials

The phenolic resins considered in this work were a resol (black resin) and a novolac (chromium platable resin) grade, utilized for the production of knobs for household appliances through injection moulding by Nuova Saimpa srl (Borgo Chiese, Italy). These materials were supplied by Nuova Saimpa srl in the form of granules, their commercial names and physical properties cannot be disclosed for confidentiality reasons. The production scrap to be used in the recycling process was provided by the same company and consisted of injection sprues (Figure 1) and scraps from the quality check of knobs. In the case of novolac-based scraps, both uncoated and chromium-based coated knobs were provided by the company.

### 2.2. Samples Preparation

Phenolic resin scrap was first ground using a Piovan^®^ RN166/1 granulator (Piovan S.p.A., Santa Maria di Sala, Italy) to a size of 2–3 mm and then further milled to dimensions lower than 60 µm using an IKA M20 Universal Mill (IKA Werke, Staufen im Breisgau, Germany). The choice of the recyclate granulometry was optimized upon preliminary trials. The recycling process was performed at room temperature to avoid any possible degradation of the material caused by overheating.

Recycled powders and virgin material were manually mixed and then melted and compounded through an internal mixer (Thermo Haake Rheomix^®^ 600, Thermo Fisher Scientific, Waltham, MA, USA), equipped with counter-rotating rotors operating at 100 °C for a time of 3 min. The compound was then ground again to a size of 2–3 mm and the granules were used to fill a mould that was hot-pressed at 24.5 bar and a temperature of 180 °C for 5 min using a Carver hot-plate press (Carver Inc., Wabash, IN, USA). In this way, square sheets (110 × 110 × 4 mm^3^) were obtained. The produced sheets were then milled using a 3-axis CNC milling machine (GP project Snc, Valsamoggia, Italy) in order to obtain regular samples for mechanical tests (80 × 10 × 4 mm^3^). The production process is schematized in Figure 2.

Table 1 reports the list of the prepared samples together with their codes: NOV and RES refer to the material used (novolac and resol, respectively), while the numbers refer to the amount of recycled material added to the composition). In Figure 3 the samples after compression moulding (a) and after CNC milling (b) are shown. Regarding the use of novolac resins, it should be pointed out that, only in one case (i.e., NOV30CR sample), the recycled material obtained from chromium-coated materials was used.

### 2.3. Methods

#### 2.3.1. Lab Scale Characterization

The cryofractured surfaces of the samples at different recyclate amounts were observed through a Jeol IT300 scanning electron microscope (JEOL Ltd., Tokyo, Japan) operating at an acceleration voltage of 10 kV. Before the observations, the specimens were metalized under vacuum through the deposition of a thin electrically conductive Pt/Pd coating.

Thermogravimetric analysis (TGA) was performed through a Mettler TG50 thermobalance (Mettler-Toledo International Inc., Columbus, OH, USA) under an air flow of 100 mL/min in a temperature interval between 30 and 800 °C, at a heating rate of 10 °C/min. The temperature associated with a mass loss of 5% (T_5%_), the temperature associated with the maximum rate of degradation (T_peak_) and the residual mass at 800 °C (m_800_) were determined.

Differential scanning calorimetry (DSC) tests were performed only on virgin materials (NOV, RES) before curing (i.e., as received) and on the production scraps (sprue, knob). Tests were performed using a Mettler DSC30 calorimeter (Mettler-Toledo International Inc., Columbus, OH, USA) at a testing speed of 10 °C/min under a nitrogen flow of 100 mL/min and consisted of a first heating scan from 0 °C to 200 °C, followed by a cooling scan from 200 °C to 0 °C and by a second heating scan from 0 °C to 200 °C. The glass transition temperature (T_g_) and the crosslinking temperature in the first heating scan (T_crl_) were determined.

Vicat softening temperature (VST) was determined following ASTM D1525 standard [26] at a heating rate of 120 °C/h using a microprocessor HDT-Vicat tester model MP/3 (ATS Faar Industries Srl, Milano, Italy) subjecting small bars of 4 mm thickness to a load of 50 N imposed by an indenter with an area of 1 mm^2^. Three specimens were tested for each composition. Shore-D hardness was determined at 25 °C following ASTM D2240 standard [27] through a Hildebrand durometer applying a load of 4 kg for 5 s, performing 10 indentations for each sample.

Three-point bending tests were performed under quasi-static conditions using an Instron 5969 tensile testing machine (Instron, Norwood, OH, USA) equipped with a load cell of 50 kN and operating at a cross-head speed of 1.5 mm/min according to the ISO 1209-2 standard [28]. Five rectangular specimens having a width of 10 mm, a thickness of 4 mm and a total length of 80 mm were tested for each sample. A span length of 64 mm was utilized for all the specimens.

Flexural tests were repeated also on samples that were recycled multiple times: NOV30 specimens prepared according to the procedure described in Section 2.2 were reprocessed (ground, melt compounded with virgin material keeping the recycled content to 30 wt% and hot pressed) for 5 and 10 times (NOV30_5° and NOV30_10°, respectively) and specimens for the tests were obtained using the CNC milling machine previously described. Only the NOV30 sample was selected for this test due to the intermediate recycled content and due to the fact that novolac samples generally showed lower mechanical properties with respect to resol one upon recycled material addition.

#### 2.3.2. Industrial Production Trials

To verify the industrial processability of the recycled materials, at least from a qualitative point of view, production trials were performed by adding the recycled powder to the virgin materials in a reaction injection moulding process carried out in Nuova Saimpa srl plants, as shown in Figure 4. Different knobs were thus obtained, and compliance with the strict aesthetical requirements imposed for household appliances (absence of any glare or stain on the surface) was assessed. In order to satisfy the elevated aesthetical features needed for design objects such as knobs, after some preliminary tests, it was decided to produce industrial prototypes with a recyclate amount of 15 wt% (i.e., NOV15 and RES15 samples). It is important to note that, in the case of non-aesthetical components, when the satisfaction of mechanical requirements is the main issue, the amount of recyclate could be increased according to the lab tests.

#### 2.3.3. Life Cycle Assessment (LCA)

The environmental impact of the knobs produced only with virgin (NOV, RES) and with recycled materials (NOV15, RES15) was determined through life cycle assessment. The aim of the study was to quantify the potential benefits arising from the substitution of virgin materials with recycled ones and also to identify the environmental hot spots in the production process of phenolic resin knobs. The analysis was carried out referring to a “cradle to grave” approach. Primary data regarding the industrial production process were provided by Nuova Saimpa srl and referred to the production of 1 year (2021), while secondary data were gathered from the database Ecoinvent 3.10. The SimaPro™ software release 9.6, supplied by Pré Sustainability (Amersfoort, Netherlands), was used for the data processing.

According to the EN15804 standard [29], the functional unit of this study was 1 knob for household appliances aimed at the intensity regulation of fire (in case of hob) or temperature (oven).

The system boundaries of the study included the following: the extraction of raw materials (phenolic resins, additives), the transportation to the production plant of Nuova Saimpa srl, the manufacturing of the knobs (injection moulding, washing, cleaning, coating, and packaging), the packaging acquisition and waste treatment of the production scrap, the distribution and the end-of-life operations (waste transportation, treatment, and disposal). Due to the lack of data regarding the waste management of phenolic resins, it was assumed that a fraction of 37 wt% is sent to landfill and the remaining (63 wt%) to incineration. These percentages were adopted considering the Europen figures for post-consumer plastic waste management in 2021 (35 wt% recycling, 42 wt% energy recovery, and 23 wt% landfill) and setting to zero the recycling share. For the end-of-life of paper and plastic packaging, European figures referred to in the year 2021 were assumed [30,31]. For the waste transportation, it was assumed a distance of 100 km. The electricity mix used in the production plants of Nuova Saimpa srl was modelled, according to the real situation in which around 5.1% of the needs were satisfied using the photovoltaic system installed on the roof of the building and the remaining was acquired from the grid with guarantee of origin certification. A specific electricity scenario was, therefore, developed in the software considering the effective share of different energy sources (86.7% hydropower, 7.2% biofuels, 6.1% photovoltaic) and with an environmental impact in terms of greenhouse gasses emissions equal to 0.183 kg CO_2_ eq/kWh.

To model the production of resol-type phenolic resin (not present in the Ecoinvent database), the original process for the production of a novolac-type phenolic resin (*Phenolic resin {RER}|phenolic resin production|Cut-off*, *U*) was modified setting a P/F ratio equal to 0.487 as reported by Solyman et al. [32].

The environmental impact assessment has been performed using the EF 3.1 methodology and considered eight impact categories: acidification, climate change, freshwater ecotoxicity, particulate matter, eutrophication (marine, freshwater, and terrestrial), human toxicity (cancer and non-cancer), ionizing radiation, land use, ozone depletion, photochemical ozone formation, resource use (fossil, mineral and metals), water use.

The main inputs of the life cycle inventory are summarized in Table 2 (the complete life cycle inventory cannot be disclosed for confidential reasons.

## 3. Results and Discussion

### 3.1. Lab Scale Characterization

The prepared samples were observed through scanning electron microscopy (SEM) in order to investigate their morphological features and to assess the effectiveness of the incorporation of recycled material within the virgin resin. From the SEM micrographs of virgin samples reported in Figure 5a,c, it is possible to observe that they are characterized by the same morphology, without porosities or inhomogeneities. On the other hand, samples containing recycled material (Figure 5b,d) present higher surface roughness and some small pores or defects and inhomogeneities, although the interfacial adhesion with recycled particles seems very good. Sample NOV30CR clearly shows the presence of coating particles of large dimensions not encapsulated within the virgin matrix and probably acting as defects, potentially able to decrease the failure properties of the resulting materials.

Looking at the thermogravimetric analysis curves shown in Figure 6a,b and the results listed in Table 3, it can be observed that virgin samples are stable up to 300 °C (RES) and 350 °C (NOV). Above these temperatures the virgin samples face three different degradation steps: the first, at around 350–370 °C (T_peak1_), corresponded to oxidative degradation; the second, at about 450–470 °C (T_peak2_), correlated to thermal fragmentation with the formation of phenol, cresole and xylenol; and the third, at around 560–600 (T_peak3_), related to the formation of carbon char and release of H_2_, CO and CH_4_ [33,34,35,36]. The small peak at around 300 °C detectable both in RES and NOV samples is not associated with a degradation of the material but to posturing of residual uncured fractions of the resin related to an incomplete curing reaction that occurred during the production process [35]. From the TGA curves, considering the residual mass at 300 °C, it is possible to roughly estimate the uncured fraction of phenolic resin: it corresponds to around 3 wt% in the case of NOV and 7 wt% in the case of RES. For samples filled with recycled material, it is possible to observe that, both for novolac and resol resin, the curves are practically overlapped up to 600 °C, meaning that no substantial changes in thermal degradation stability occur. The char residue at 800 °C (m_800_) is slightly higher in the case of novolac samples (~20 wt%), while in the case of resol samples, it is around 10 wt%, independently of recycled content.

From the first scan of DSC curves, shown in Figure 7a,b, it is possible to identify, on the curves of the uncured sample, an inflexion point at around 60 °C, corresponding to the glass transition (T_g_) of the phenolic resin [37]. According to the results reported in Table 4, it is slightly lower for the novolac resin (57 °C), with respect to the resol one (63 °C). At higher temperatures, only for the NOV sample it is possible to observe an exothermic peak related to the crosslinking reaction of the material, that reaches its maximum at around 159 °C (T_crl_). In the case of the RES sample, it is possible to identify an endothermic peak at around 130 °C correlated to the release of water (co-product of the condensation reaction between phenol and aldehyde), followed by a small exothermic peak (165 °C) related to the crosslinking of the resin [38]. In the case of crosslinked materials (sprues and knobs) produced with novolac resin, it is possible to identify only a small shoulder at around 110 °C, which is more evident in the case of materials produced starting from resol: this behaviour can be related to the crosslinking reaction of uncured residues, as also observed in the thermogravimetric analyses. This crosslinking residue could affect the mechanical behaviour of the samples containing recycled material.

From the cooling and the second heating scan thermograms, it is not possible to distinguish any peak, as the materials are completely crosslinked.

The results of Vicat tests reported in Table 5 show that the dimensional stability of the prepared novolac samples is almost unaffected by the addition of recycled material: the Vicat softening temperature (VST) is above the upper limit detectable by the instrument (280 °C) for all the samples. Moreover, also the penetration value at 280 °C seems to be not influenced by the recyclate addition, with the exception of NOV30CR that shows higher penetration values with respect to other samples, probably due to the interfacial weakening effect played by the chromium-based coating present on the recycled particles. Resol samples, as also observed from TGA, present lower thermal resistance highlighted by the higher penetration values reached at 280 °C (up to 1 mm for RES40). Also, Shore D hardness values, performed at room temperature, show that the addition of recycled material does not affect the indentation resistance of the material.

Quasi-static three-point bending tests were carried out to detect the influence of recycled material addition on the mechanical properties of the produced samples. From the flexural stress–strain curves shown in Figure 8a,b and from the results reported in Table 6, it is possible to observe that the recyclate addition has almost no influence on the mechanical properties of resol-based samples, which show the same flexural modulus, strength and strain at break of the virgin resin. In the case of novolac-based samples, it is possible to notice a limited decrease in the mechanical properties upon recycled material addition, and this drop is more evident at high recycled content (flexural strength decreases of 20% in case of NOV40) and for samples produced by using chromium-plated recycled material (NOV30CR) that shows a flexural strength decrease of around 27%. The different trend highlighted by novolac and resol samples upon recycled material addition may be related to the different uncured resin amounts that were detected both in TGA and DSC tests: the higher uncured resin amount observed for the RES samples (doubled with respect to NOV samples) is probably the reason behind the higher mechanical properties observed also upon recyclate addition. On the other hand, the decrease in the failure properties noticed for NOV30CR sample is probably related to the presence of chromium coating that, as observed in Figure 5b, presents a weak interfacial adhesion with the phenolic resin matrix.

To evaluate the effect of multiple recycling cycles on the mechanical properties of novolac and resol samples, three-point bending tests on a selected composition (NOV30) containing a constant amount (30wt%) of recycled material obtained after 5 and 10 reprocessing cycles were performed. From the flexural stress–strain curves shown in Figure 9, it is possible to observe a modest decrease in the mechanical properties upon multiple recycling stages. Looking at the results listed in Table 7, it is possible to quantify, with respect to the virgin material (NOV sample), a decrease in the flexural modulus of 4% and 9%, in the flexural strength of 33% and 41% and in the strain at break of 31% and 35% for NOV30_5° and NOV30_10° samples, respectively. These drops, higher with respect to those recorded for the NOV30 sample (−4%, −13% and −11%) are probably caused by the fact that, starting from the second recycling cycle, the material loses any residual uncured content and behaves as a completely inert material. Moreover, the further mechanical performance drop recorded after multiple recycling cycles may be related to the partial degradation of the resin caused by the reprocessing operations [17]. However, it can be concluded that the observed losses are not dramatic, and the mechanical features of the reprocessed material are suitable for the industrial production of components with the same application as the original ones, even after 10 recycling stages.

### 3.2. Industrial Production Trials

Industrial production trials were performed using the same procedure and the same injection moulding instrumentation used for virgin materials. The essential requirement to be verified was the attainment of the strict aesthetical requirements (no differences with respect to the original components). For this reason, after preliminary tests, it was decided to use a lower recyclate amount with respect to lab tests (15 wt%): at higher contents, despite the functionality being guaranteed, the presence of glare or stain could be detected. As it is possible to observe from the representative pictures reported in Figure 10a,b), both for novolac and resol samples it is not possible to distinguish between the virgin sample and the one containing recycled material, at least with a recyclate amount of 15 wt%. This qualitatively confirms the feasibility of the mechanical recycling process at the industrial level, giving the possibility of a complete re-introduction of the production scraps within the industrial cycle.

Future work, on this side, will include the determination of the maximum recycled content for each knob geometry, since different shapes require different injection moulding parameters with consequent different levels of maximum recycled content that could be added in the composition, in order to guarantee the satisfaction of the aesthetical requirements.

### 3.3. Life Cycle Assessment

The third part of the work regarded the quantification of the potential environmental benefits arising from the recycling process and was performed through a Life Cycle Assessment of both NOV and RES knobs, performed using primary data referring to the industrial production process. The main inputs used for the life cycle inventory are reported in Table 2.

The absolute values of the environmental impact of NOV and NOV15 knobs are presented in Table 8. Observing these results, it is possible to notice that the use of recycled material allows a modest decrease in all the impact categories of around −2/3%. To understand the limited benefit arising from the addition of 15 wt% of recycled material it is necessary to observe Figure 11, which shows the relative contribution of the different production stages to the overall environmental impact. Taking the impact category *Climate change* as a reference, it is possible to observe that only 15% of the environmental impact is caused by the raw materials, while more than 50% is attributed to the chromium coating applied on the sample surface for aesthetical reasons. Looking at the environmental impact of RES and RES15 knobs, whose values are reported in Table 9, it is possible to observe that the substitution of 15 wt% of virgin material with the recycled one allows a general decrease of around −7/10% on all impact categories. This different behaviour with respect to novolac knobs can be understood by looking at Figure 11b: in this case, since resol knobs are not chromium coated, the environmental impact is mainly caused by the raw materials that, in the *climate change* impact category, are responsible for more than 40% of the overall impact. Comparing the NOV and RES knobs, it can be also noted that, despite the lower mass of NOV knobs (total mass of 15.3 and 51.8 g, respectively), it is characterized by a very high environmental impact (considering that the mass is around three times lower the expected *climate change* value, it should be around 13 g CO_2_ eq instead of 36 g CO_2_ eq). This difference is mainly caused by the fact that NOV knobs are subjected to a chromium plating process, while RES knobs are only to an acrylic painting with lower thickness (the amount of material deposited is around 1.3 g in the case of NOV knob and 0.2 g in case of RES) and also due to the fact that novolac resin has a higher environmental impact with respect to resol due to the higher phenol content. In conclusion, the Carbon Footprint indicator, corresponding to the impact category *Climate change*, is equal to 36.3 g CO_2_ eq. for 1 NOV knob and 47.3 g CO_2_ eq. for 1 RES knob. These values can be reduced to −3.8 and −8.1% upon substitution of 15 wt% of virgin material with recycled one within the production process.

The group analysis shown in Figure 11a,b is a useful instrument in order to identify possible strategies for the reduction in the environmental impact of the two products. In the case of the production of novolac knobs, the only way to obtain a substantial decrease in the environmental impact is the identification of alternatives to the current chromium coating process. It should be noted that the high environmental impact in the *Water use* category is related to the use of hydropower as the main source of the energy mix used by the company and not by the use of water in the production process. In the case of resol-based knobs, the reduction in the environmental impact is possible through the use of recycled material (that also has the advantage of reducing the high impact of the waste treatment) and through the optimization of the production process, with a consequent reduction in energy consumption.

## 4. Conclusions

In the present work, various amounts of both novolac and resol phenolic resins obtained from production scrap were mixed with virgin material and used for the production of knobs for household appliances. From the characterization of samples produced at lab scale, it was observed that the thermal properties were not influenced by the presence of recycled material, while from the mechanical tests, it was noticed that amounts up to 30 wt% of recycled material could be added to the virgin resins without significant impairment of flexural properties. The production trials performed at an industrial scale highlighted the feasibility of adding 15 wt% of recycled material, without affecting the production process and/or compromising the aesthetical quality of the produced knobs. Finally, a comparative life cycle assessment performed to quantify the benefits arising from the partial substitution of virgin material with recycled one highlighted a limited benefit in the case of novolac base knobs, due to the high impact of the chromium coating applied for aesthetical reasons. In the case of resol-based knobs, a higher decrease in the environmental impact (around 7–10% in all impact categories) was highlighted. Considering the very promising results of this study, further efforts will be focused on the identification of the maximum recyclate content achievable for each knob configuration produced by the industrial partner and the identification of more environmentally friendly alternatives to the chromium plating process.

## Figures and Tables

**Figure 1 polymers-16-03378-f001:**
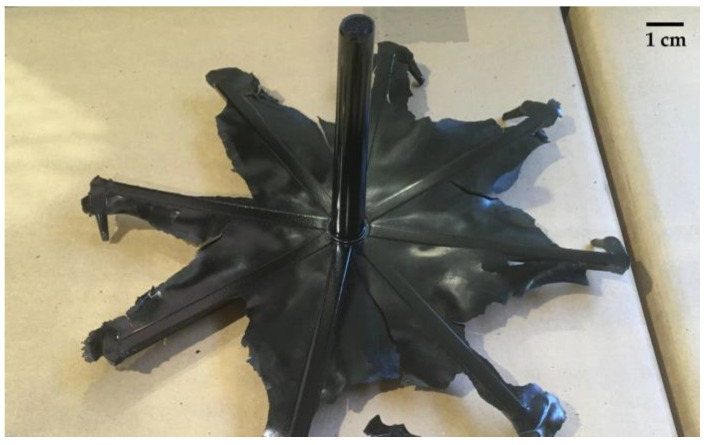
Example of injection moulding sprue used as recyclate in the mechanical recycling process.

**Figure 2 polymers-16-03378-f002:**
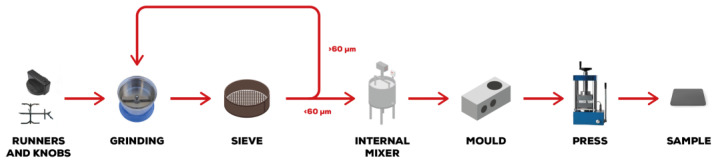
Production scheme for the samples tested at lab scale.

**Figure 3 polymers-16-03378-f003:**
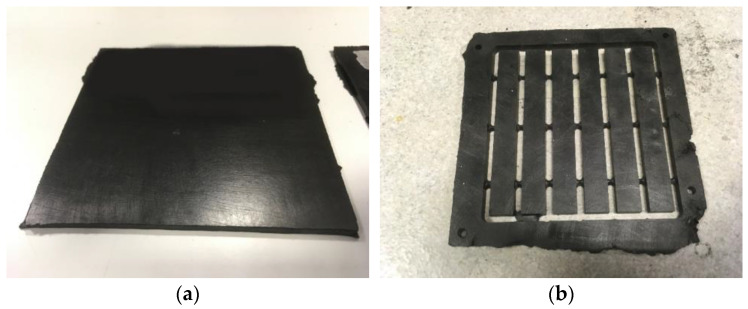
Pictures of the prepared samples (NOV30) using recycled material: (**a**) after compression moulding, and (**b**) after CNC milling.

**Figure 4 polymers-16-03378-f004:**
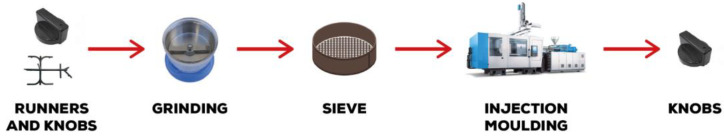
Production scheme for the industrial production trials.

**Figure 5 polymers-16-03378-f005:**
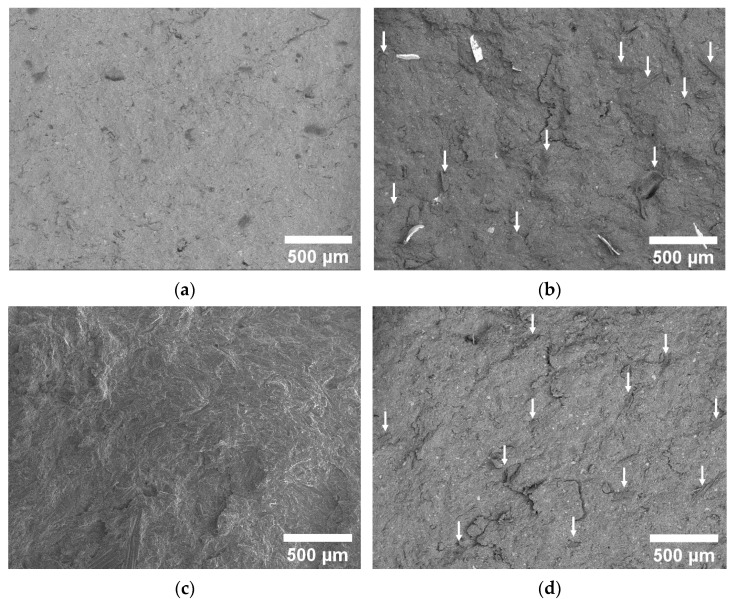
SEM micrographs of the cryofractured surface of (**a**) NOV, (**b**) NOV30CR, (**c**) RES, and (**d**) RES30 samples. Recycled particles are highlighted by arrows.

**Figure 6 polymers-16-03378-f006:**
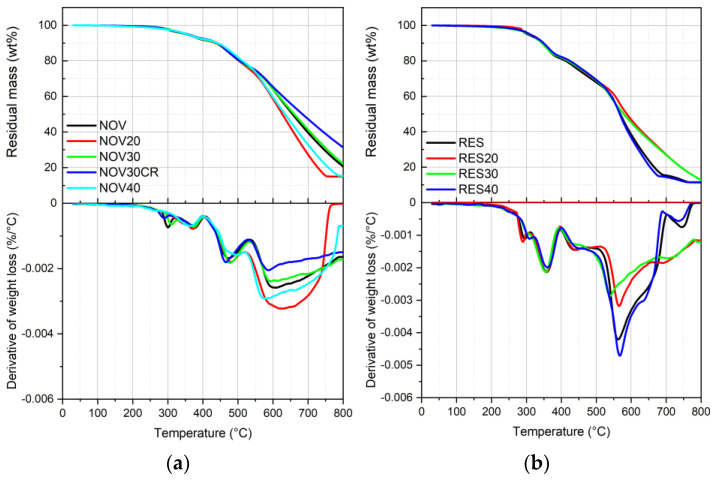
TGA curves of (**a**) novolac and (**b**) resol samples containing different amounts of recycled material.

**Figure 7 polymers-16-03378-f007:**
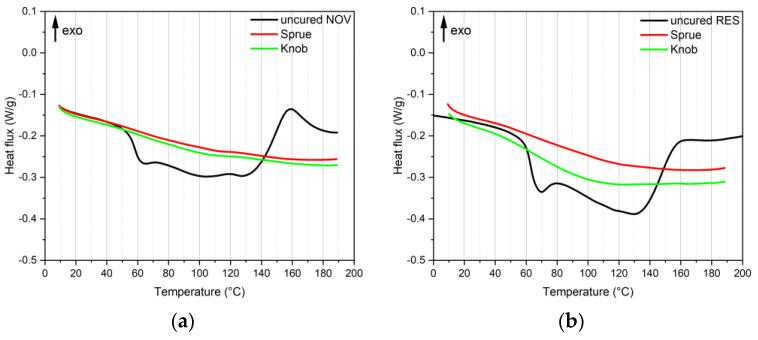
DSC curves (first heating scan) of novolac (**a**) and resol (**b**) samples.

**Figure 8 polymers-16-03378-f008:**
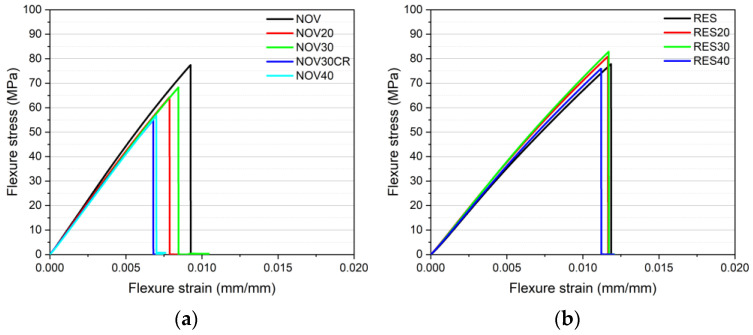
Representative flexural stress–strain curves of (**a**) novolac and (**b**) resol-based samples containing different amounts of recycled material.

**Figure 9 polymers-16-03378-f009:**
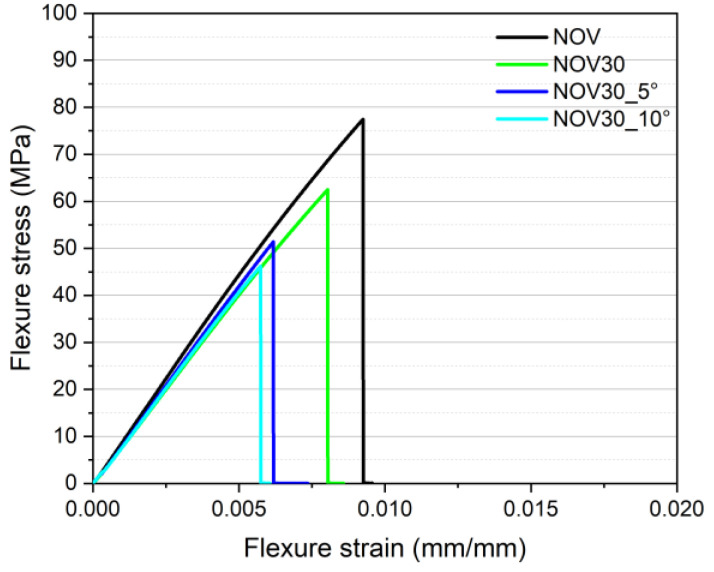
Flexural stress–strain curves of novolac samples produced using recycled material subjected to multiple reprocessing cycles.

**Figure 10 polymers-16-03378-f010:**
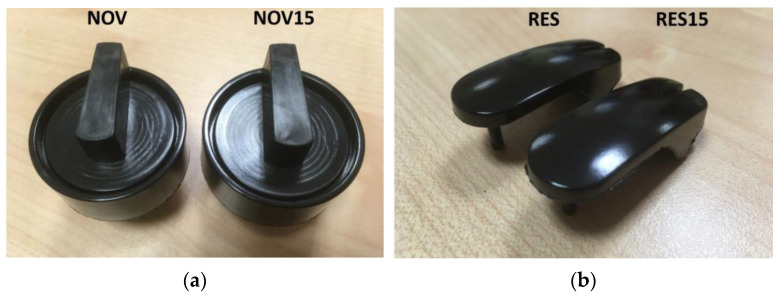
Representative images of knobs produced upon industrial production trials based on (**a**) NOV and NOV15 samples, and (**b**) RES and RES15 samples.

**Figure 11 polymers-16-03378-f011:**
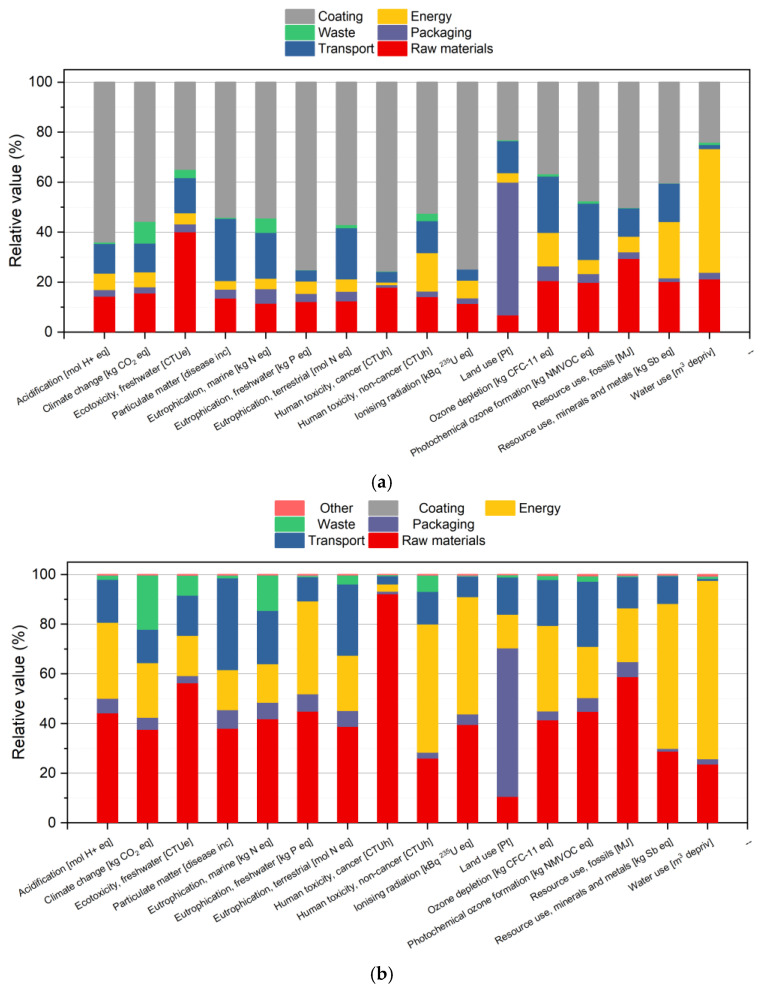
Group analysis on the life cycle impact assessment results of the industrial production of a knob based on (**a**) NOV and (**b**) RES samples.

**Table 1 polymers-16-03378-t001:** List of the prepared samples.

Sample	Virgin [wt%]	Recycled [wt%]
NOV	100	0
NOV20	80	20
NOV30	70	30
NOV30CR *	70	30
NOV40	60	40
RES	100	0
RES20	80	20
RES30	70	30
RES40	60	40

* represents preparation using only chromium-plated recycled material.

**Table 2 polymers-16-03378-t002:** Main inputs of the life cycle inventory used in the life cycle assessment of knobs produced only with virgin (NOV, RES) and with the addition of recycled resin (NOV15, RES15).

	NOV	NOV15	RES	RES15
**Final mass [g]**	15.31		51.8	
**Recycled content [wt%]**	0	15	0	15
**Electricity consumption (total) [MJ]**	0.39	0.39	1.87	1.87
**Transports (supply) ^1^ [kg** × **km]**	17.60	17.60	52.10	52.10
**Transports (distribution) ^2^ [kg** × **km]**	11.50	11.50	12.60	12.60
**Surface treatment ^3^**	Cromium coating 1.34 g		Acrylic paint 0.20 g	
**Plastic packaging [g]**	0.15	0.15	0.20	0.20
**Paper packaging [g]**	5.50	5.50	3.50	3.60

^1^ Supply transports modelled according to the primary data provided by Nuova Saimpa srl using the Ecoinvent process: *transport*, *freight*, *lorry 3.5–7.5 metric ton*, *EURO6 {RER} market for transport*, *freight*, *lorry 3.5–7.5 metric ton*, *EURO6|Cut-off*, *S*. ^2^ Distribution transports modelled according to the primary data provided by Nuova Saimpa srl using the Ecoinvent process: *transport*, *freight*, *light commercial vehicle {Europe without Switzerland}|market for transport*, *freight*, *light commercial vehicle|Cut-off*, *S*. ^3^ The production process of novolac knobs includes a surface treatment performed through a chemical and electrolytic plating that involves first the deposition of a Nichel layer as substrate to promote the adhesion of the second layer made of Chromium. This process is performed in an external factory and, due to the unavailability of primary data, it was modelled using the Ecoinvent process “*Hard chromium coat*, *electroplating*, *steel substrate*, *0.14 mm thickness {GLO}|market for hard chromium coat*, *electroplating*, *steel substrate*, *0.14 mm thickness|Cut-off*, *S*”.

**Table 3 polymers-16-03378-t003:** Results of TGA tests on the prepared samples.

Sample	T_5%_ [°C]	T_peak1_ [°C]	T_peak2_ [°C]	T_peak3_ [°C]	m_800_ [wt%]
**NOV**	351.3	375.5	473.7	604.0	20.7
**NOV20**	353.8	370.8	479.8	619.8	14.9
**NOV30**	353.5	373.3	477.3	592.2	22.1
**NOV30CR**	356.7	370.5	466.2	587.3	31.4
**NOV40**	354.2	361.2	484.4	572.2	15.3
**RES**	303.7	358.0	437.0	562.8	11.4
**RES20**	309.3	355.7	441.3	565.0	12.6
**RES30**	306.2	354.8	430.3	539.0	12.6
**RES40**	294.2	360.6	449.2	568.2	11.5

**Table 4 polymers-16-03378-t004:** Results of the first heating scan of DSC tests on the uncured materials.

Sample	T_g_ [°C]	T_crl_ [°C]
**uncured NOV**	57.2	158.8
**uncured RES**	63.2	165.9

**Table 5 polymers-16-03378-t005:** Results of Vicat and of Shore-D hardness tests on the prepared samples.

Sample	Vicat Softening Temperature (VST) [°C]	Penetration at 280 °C [mm]	Shore D
**NOV**	>280	0.10 ± 0.17	91 ± 1
**NOV20**	>280	0.15 ± 0.07	91 ± 1
**NOV30**	>280	0.18 ± 0.03	91 ± 1
**NOV30CR**	>280	0.33 ± 0.07	91 ± 1
**NOV40**	>280	0.13 ± 0.11	91 ± 1
**RES**	>280	0.82 ± 0.04	90 ± 2
**RES20**	>280	0.72 ± 0.14	91 ± 1
**RES30**	>280	0.79 ± 0.02	91 ± 1
**RES40**	280	1.00 ± 0.01	90 ± 1

**Table 6 polymers-16-03378-t006:** Results of three-point bending tests on the prepared samples.

Sample	Flexural Modulus [GPa]	Flexural Strength [MPa]	Flexural Strain [%]
**NOV**	8.9 ± 0.1	74.1 ± 3.6	0.89 ± 0.05
**NOV20**	8.5 ± 0.1	61.8 ± 3.0	0.76 ± 0.04
**NOV30**	8.5 ± 0.1	64.5 ± 8.5	0.79 ± 0.11
**NOV30CR**	8.5 ± 0.1	54.1 ± 5.4	0.65 ± 0.07
**NOV40**	8.3 ± 0.2	59.3 ± 3.8	0.74 ± 0.08
**RES**	7.3 ± 0.2	76.2 ± 4.7	1.15 ± 0.10
**RES20**	7.5 ± 0.2	74.7 ± 11.5	1.08 ± 0.19
**RES30**	7.5 ± 0.2	81.4 ± 5.6	1.19 ± 0.12
**RES40**	7.5 ± 0.2	77.0 ± 0.9	1.11 ± 0.03

**Table 7 polymers-16-03378-t007:** Results of three-point bending tests on the novolac samples produced using recycled material subjected to multiple reprocessing cycles.

Sample	Flexural Modulus [GPa]	Flexural Strength [MPa]	Flexural Strain [%]
**NOV**	8.9 ± 0.1	74.1 ± 3.6	0.89 ± 0.05
**NOV30**	8.5 ± 0.1	64.5 ± 8.5	0.79 ± 0.11
**NOV30_5°**	8.5 ± 0.1	49.8 ± 1.8	0.61 ± 0.02
**NOV30_10°**	8.1 ± 0.5	43.2 ± 3.3	0.58 ± 0.06

**Table 8 polymers-16-03378-t008:** Results of the comparative life cycle impact assessment related to the production of 1 knob based on NOV and NOV15 samples.

Impact Category	Unit	NOV	NOV15	Δ [%]
Acidification	mol H^+^ eq	1.58 × 10 × 10^−3^	1.55 × 10^−3^	−2.4
Climate change	kg CO_2_ eq	3.63 × 10^−1^	3.49 × 10^−1^	−3.8
Ecotoxicity, freshwater	CTUe	2.38 × 10^0^	2.17 × 10^0^	−8.8
Particulate matter	disease inc.	1.69 × 10^−8^	1.64 × 10^−8^	−2.5
Eutrophication, marine	kg N eq	3.42 × 10^−4^	3.33 × 10^−4^	−2.6
Eutrophication, freshwater	kg P eq	1.12 × 10^−4^	1.10 × 10^−4^	−2.1
Eutrophication, terrestrial	mol N eq	3.40 × 10^−3^	3.32 × 10^−3^	−2.3
Human toxicity, cancer	CTUh	2.51 × 10^−9^	2.29 × 10^−9^	−8.7
Human toxicity, non-cancer	CTUh	2.94 × 10^−9^	2.83 × 10^−9^	−3.6
Ionizing radiation	kBq U-235 eq	3.43 × 10^−2^	3.37 × 10^−2^	−1.9
Land use	Pt	2.05 × 10^0^	2.00 × 10^0^	−2.1
Ozone depletion	kg CFC11 eq	4.97 × 10^−9^	4.66 × 10^−9^	−6.3
Photochemical ozone formation	kg NMVOC eq	1.28 × 10^−3^	1.23 × 10^−3^	−3.7
Resource use, fossils	MJ	5.17 × 10^0^	4.92 × 10^0^	−4.7
Resource use, minerals and metals	kg Sb eq	2.24 × 10^−6^	2.14 × 10^−6^	−4.2
Water use	m^3^ depriv.	1.32 × 10^−1^	1.30 × 10^−1^	−1.3

**Table 9 polymers-16-03378-t009:** Results of the comparative life cycle impact assessment related to the production of 1 knob based on RES and RES15 samples.

Impact Category	Unit	RES	RES15	Δ [%]
Acidification	mol H^+^ eq	1.63 × 10^−3^	1.52 × 10^−3^	−6.2
Climate change	kg CO_2_ eq	4.73 × 10^−1^	4.35 × 10^−1^	−8.1
Ecotoxicity, freshwater	CTUe	3.14 × 10^0^	2.68 × 10^0^	−14.7
Particulate matter	disease inc.	1.74 × 10^−8^	1.63 × 10^−8^	−6.2
Eutrophication, marine	kg N eq	4.43 × 10^−4^	4.17 × 10^−4^	−5.9
Eutrophication, freshwater	kg P eq	7.31 × 10^−5^	6.67 × 10^−5^	−8.8
Eutrophication, terrestrial	mol N eq	3.69 × 10^−3^	3.46 × 10^−3^	−6.1
Human toxicity, cancer	CTUh	4.54 × 10^−9^	3.75 × 10^−9^	−17.3
Human toxicity, non-cancer	CTUh	4.35 × 10^−9^	4.05 × 10^−9^	−7.0
Ionizing radiation	kBq U-235 eq	2.47 × 10^−2^	2.25 × 10^−2^	−9.1
Land use	Pt	3.50 × 10^0^	3.35 × 10^0^	−4.1
Ozone depletion	kg CFC11 eq	1.02 × 10^−8^	8.82 × 10^−9^	−13.2
Photochemical ozone formation	kg NMVOC eq	1.66 × 10^−3^	1.51 × 10^−3^	−8.9
Resource use, fossils	MJ	7.27 × 10^0^	6.51 × 10^0^	−10.5
Resource use, minerals and metals	kg Sb eq	3.93 × 10^−6^	3.65 × 10^−6^	−7.1
Water use	m^3^ depriv.	4.83 × 10^−1^	4.78 × 10^−1^	−1.0

## Data Availability

The data presented in this study are available on request from the corresponding author due to confidentiality reasons.

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
