# Peer review of "Optimization of the Mechanical Recycling of Phenolic Resins for Household Appliances"

_polymers, 2024, doi:10.3390/polym16233378_

Round 1

Reviewer 1 Report

Comments and Suggestions for Authors

The authors studied the mechanical recycling obtained from phenolic resin waste and optimized it by partially replacing the original material. The feasibility of production was studied on both laboratory and industrial scale. The environmental benefits are also evaluated. This work is very meaningful, but there are some problems that need to be solved.

1. A simple and clear diagram is necessary, which helps to improve the reader's understanding by introducing the specific process.

2. Section 2.2:The phenolic resin waste was ground several times. Does this result in the release of phenolic substances or free formaldehyde? What is the release? This may cause "secondary pollution" to the environment.

3. L159:The author chose 15wt% as the additive amount in order to satisfy the aesthetic characteristics. However, this sample is not the best. This puzzled me. Can the best sample not achieve aesthetic satisfaction? Why can't it happen? The author should analyze this.

4. There are defects in the samples containing the recovered material, which can lead to a deterioration of their mechanical properties. Is this due to a simple grinding and heating process? Mixing at the molecular level may help to improve the situation.

5. Standard formats are required, e.g. selection of significant figures, etc.

Comments on the Quality of English Language

English editing could be improved.

Author Response

The authors studied the mechanical recycling obtained from phenolic resin waste and optimized it by partially replacing the original material. The feasibility of production was studied on both laboratory and industrial scale. The environmental benefits are also evaluated. This work is very meaningful, but there are some problems that need to be solved.

  1. A simple and clear diagram is necessary, which helps to improve the reader's understanding by introducing the specific process.

The authors thank the reviewer for the suggestion. Two diagrams were added for clarity.

  1. Section 2.2:The phenolic resin waste was ground several times. Does this result in the release of phenolic substances or free formaldehyde? What is the release? This may cause "secondary pollution" to the environment.

According to TGA analysis, below 200 °C there is no mass loss, this means that since the grinding process is performed at room temperature we should not have the emission of any component. The reaction of unreacted components could occur during the reprocessing but this is exactly the same that happens during the first processing step. Regarding the release of free formaldehyde the produced components already satisfy the new limits fixed by EC (0.062 mg/m3) and the eventual emission of free formaldehyde during the grinding process would in any case occur during the final waste treatment also for components that are not subjected to the recycling process. In this sense it seems to the authors that the problem of free formaldehyde is not related to the recycling process but the production process, that is not modified in this work with respect to the standard one.

  1. L159:The author chose 15wt% as the additive amount in order to satisfy the aesthetic characteristics. However, this sample is not the best. This puzzled me. Can the best sample not achieve aesthetic satisfaction? Why can't it happen? The author should analyze this.

The authors thank the reviewer for the valuable and interesting comment. The best samples (from a recycled content point of view) are NOV40 and RES40, the main problem related to the industrial production process is the fact that at this recycled content the produced samples present small defects (glare, stain) that do not impair their functionality but make them unacceptable from the aesthetical point of view due to the fact that they need to be perfect, without any difference with respect to the original ones. These aspects were better clarified in the manuscript.

  1. There are defects in the samples containing the recovered material, which can lead to a deterioration of their mechanical properties. Is this due to a simple grinding and heating process? Mixing at the molecular level may help to improve the situation.

The authors thank the reviewer for the suggestion. The presence of defects in the samples containing recycled material is related to the incorporation of recycled particles (with limited compatibility due to their inert nature after curing) within the virgin matrix. Mixing at the molecular level may be a good suggestion for future work but was not feasible in this work due to industrial reasons that limited the possible operations on the production scrap only to grinding and mixing with the virgin resin.

  1. Standard formats are required, e.g. selection of significant figures, etc.

The authors thank the reviewer for the comment. The application of standard formats was verified across the manuscript: Table 2 was corrected accordingly. The different format used for Table 8 and 9 is necessary due to the different order of magnitude of the selected impact categories (from 10-9 for human toxicity up to 100 for Ecotoxicity).

Reviewer 2 Report

Comments and Suggestions for Authors

In this work, the mechanical recycling of phenolic resins from industrial production scrap of plastic knobs for household appliances and the feasibility of a partial substitution of virgin materials with the recycled resins were investigated both at a laboratory and industrial scale. At the same time, the environmental benefits arising from the use of recycled material were quantified through a life cycle assessment (LCA). The results demonstrated that the recycled phenolic resins could be used instead of partial virgin phenolic resins for the manufacture of products such as knobs. It is a good example for the recycling of the phenolic resins. Main comments and suggestions are as follows:

1)     In the manuscript, the samples names like NOV, NOV 20, NOV 40; RES, RES20,RES 40 should be clearly defined.

2)     Why did authors use different amount of recycled phenolic resins in samples for the characterization on a laboratory and industrial scale?

3)     Why is the change tendency of the flexural properties of NOV series resins different from that of RES series resins as shown in Table 6?

4)     The writing format for reference 34 is different from other references. There is a typos “Figure 10” in line 374 on page 12 and “LCIA” in line 411 on page 14.

5)     Is it possible that authors discuss more on the scientific level in the manuscript?

Author Response

In this work, the mechanical recycling of phenolic resins from industrial production scrap of plastic knobs for household appliances and the feasibility of a partial substitution of virgin materials with the recycled resins were investigated both at a laboratory and industrial scale. At the same time, the environmental benefits arising from the use of recycled material were quantified through a life cycle assessment (LCA). The results demonstrated that the recycled phenolic resins could be used instead of partial virgin phenolic resins for the manufacture of products such as knobs. It is a good example for the recycling of the phenolic resins. Main comments and suggestions are as follows:

  • In the manuscript, the samples names like NOV, NOV 20, …NOV 40; RES, RES20,…RES 40 should be clearly defined.

The authors thank the reviewer for the comment. A brief explanation of the samples names was added to the manuscript.

  • Why did authors use different amount of recycled phenolic resins in samples for the characterization on a laboratory and industrial scale?

On laboratory scale we assessed the maximum amount of recycled material that could be added to the material, on the other hand on industrial scale we faced the problem of aesthetical requirements (difficult to be considered at lab scale) and this drastically reduced the amount of recycled material that could be added due to the formation of small defects that are unacceptable for the commercial products. This aspect, already present in manuscript, was clarified according to the comment.

  • Why is the change tendency of the flexural properties of NOV series resins different from that of RES series resins as shown in Table 6?

The authors thank the reviewer for the valuable comment. The main idea is that the different tendency of flexural properties between NOV and REV is mainly related to the presence of certain amount of uncured resin in recycled material used for the production of REV samples. The presence of unreacted material was hypothesized according to DSC and TGA tests. These aspects, already mentioned in the manuscript, were clarified.

  • The writing format for reference 34 is different from other references. There is a typos “Figure 10” in line 374 on page 12 and “LCIA” in line 411 on page 14.

The references were carefully revised and corrected, typos were also corrected.

5)     Is it possible that authors discuss more on the scientific level in the manuscript?

The authors thank the reviewer for the comment: we tried to improve the scientific discussion in the manuscript although it is difficult to provide a specific answer or improvement since the comment is rather vague.

Round 2

Reviewer 2 Report

Comments and Suggestions for Authors

The authors have addressed my comments in report 1. No further comments for the revised manuscript.